# Fisetin Modulates Toll-like Receptor-Mediated Innate Antiviral Response in Chikungunya Virus-Infected Hepatocellular Carcinoma Huh7 Cells

**Rafidah Lani [1], Boon-Teong Teoh [2], Sing-Sin Sam [2], Sazaly AbuBakar [2,\*] and Pouya Hassandarvish [2,\*]**

[1] Department of Medical Microbiology, Faculty of Medicine, Universiti Malaya, Kuala Lumpur 50603, Malaysia

[2] Tropical Infectious Diseases Research and Education Centre (TIDREC), Universiti Malaya, Kuala Lumpur 50603, Malaysia

\* Correspondence: sazaly@um.edu.my (S.A.); pouyahassandarvish@um.edu.my (P.H.)

**Abstract:** In the chronic phase of chikungunya virus (CHIKV) infection, excessive inflammation manifests as incapacitating joint pain and prolonged arthritis. Arthritis resulted from a large influx of infiltrating immune cells driven by pro-inflammatory cytokines and chemokines originating from the toll-like receptor (TLR)-mediated innate antiviral response. This study investigated fisetin's ability to modulate TLR-mediated antiviral responses against CHIKV in Huh7 cells. The CHIKV inhibitory potential of fisetin was assessed by plaque-forming unit assay, virus yield reduction assay, and bright-field microscopy (cytopathic effect, immunofluorescence). Fisetin's modulatory potential on TLR-mediated antiviral response was evaluated by immunofluorescence assay (expression of TLR proteins), qRT-PCR (mRNA level of antiviral genes), human cytokine array, and the immunoblotting of key transcription factors. The present study showed fisetin induced the expression of the antiviral genes at an early time-point by promoting the phosphorylation of IRF3 and IRF7. Fisetin reduced excessive inflammatory cytokine responses in CHIKV-infected Huh7 cells by impeding the over-phosphorylation of NF-κB. Fisetin also reduced CHIKV-induced cytopathic effects in CHIKV-infected Huh7 cells. Altogether, our study suggests that fisetin modulates TLR-mediated antiviral responses by affecting the CHIKV-induced inflammatory responses.

**Keywords** infectious diseases; chikungunya; antiviral; flavonoids; fisetin; toll-like receptors

## 1. Introduction

Chikungunya virus (CHIKV) is one of the Old World arthritogenic alphaviruses of the *Togaviridae* family transmitted by mosquito vectors, specifically *Ae. aegypti* and *Ae. albopictus*, between vertebrate hosts [1]. Its ~11.8 kb positive-sense single-stranded RNA (ssRNA) genome bears two open reading frames (ORFs) encoding for four non-structural proteins (nsPs), three structural proteins, and three peptides enveloped in an icosahedral capsid [2]. In the acute stage of CHIKV infection, the infected person presents with a combination of symptoms, including the abrupt onset of fever, joint swelling, joint pain, muscle pain, headache, nausea, fatigue, and a rash within 2 to 12 days after the infected-mosquito bite [3]. Although most CHIKV cases will resolve in complete recovery with lifelong immunity, some CHIKV cases could progress into a chronic stage manifested as months to years of debilitating arthritis [4].

CHIKV is classified into three genotypes, namely West African (WA), Asian, and East/Central/South African (ECSA) genotypes [5]. The WA genotype is mainly associated with enzootic transmission and small focal outbreaks in the human population, while the ECSA and Asian lineages have repeatedly spread to new regions and caused significant urban epidemics [6]. The re-emergence of CHIKV in 2004 caused explosive outbreaks in

Kenya and advanced to Comoros and the La Réunion islands between 2005–2007, resulting in the emergence of an *Ae. albopictus*-adaptive mutation (E1-226V) in the Indian Ocean Lineage-IOL strains of genotype ECSA [7]. Continuous sporadic outbreaks called for a vaccine or antiviral development. Currently, only symptomatic or supportive treatments are offered to CHIKV-infected individuals, in the form of acetaminophen or paracetamol intake to relieve fever and administer non-steroidal anti-inflammatory drugs (NSAID), with plenty of rest and fluids [8].

Fisetin, a flavonol, is one of the flavonoids exhibiting in silico and in vitro anti-CHIKV [9,10]. Flavonoids are a group of plants' secondary metabolites with a chemical backbone of a fifteen-carbon skeleton consisting of two benzene rings linked via a heterocyclic pyrane ring [11]. In plants, flavonoids serve as a secondary antioxidant defense system and regulate growth factors such as auxin [12]. Apart from their antioxidant and health-promoting properties in humans, flavonoids also possess anti-inflammatory [13], spasmolytic [14], sedative [15], antiseptic [16], anti-diabetic [17], immunostimulant [18], hepatoprotective activities [19], antimicrobial [20], and antiviral properties [21].

Fisetin has been shown to possess antiviral properties, including reducing CHIKV RNA replication, cytopathic effects, and viral protein expression [10]. The antiviral mechanisms could involve toll-like receptor-mediated pathways such as inflammation, apoptosis, and autophagy [22–24]. Toll-like receptors (TLRs) are one of the pathogen recognition receptors that are most studied in innate immunity. TLRs involve directly and/or indirectly regulating inflammation [25], autophagy [26], apoptosis [27], oxidative stress [28], antiviral [29], cancer [30], antigen presentation [31], immunoglobulin (Ig)switching [32], B1 cell expansion [33], the upregulation of major histocompatibility complex [34], T cells chemotactic effects [35], co-stimulatory cytokines [36], as well as complement and coagulation cascade [37].

There is still very little known about TLR signaling pathway regulation in CHIKV infection. In mouse neuronal cells, CHIKV ECSA mutant (E1:226V) induced lesser antiviral genes and TLR3 and TLR7 compared to the novel CHIKV ECSA genotype [38]. Differential regulation in the TLR signaling pathway is attributed to increased virulence in the mutant virus. Hence, amplifying the TLR pathway before CHIKV infection might effectively control CHIKV infection as pre-treatment of mouse neuronal cells with TLR agonists; Poly I:C, IFN-$\beta$, and TNF-$\alpha$, resulting in inhibition of virus replication [38]. Single nucleotide polymorphism in TLR7 and TLR8 influenced human CHIKV-susceptibility and disease progression and is perceived as potential prognostic biomarkers for predicting susceptibility to CHIKV infection among uninfected individuals [39]. Here, we investigate the regulation of TLR-mediated signaling pathways in CHIKV-infected and fisetin-treated hepatocellular carcinoma Huh7 cells.

## 2. Materials and Methods

### 2.1. Cell Lines and Virus

Human hepatocellular carcinoma (Huh7) cells [JCRB0403] were obtained from the Japanese Collection of Research Bioresources (JCRB) Cell Bank of National Institutes of Biomedical Innovation, Health and Nutrition (NIBIOHN, Osaka, Japan). Huh7 cells were maintained in maintenance media consisting of Roswell Park Memorial Institute, RPMI-1640 medium (ATCC® 30-2001™, Manassas, VA, USA) supplemented with 10% fetal bovine serum (FBS) (Gibco®, Palo Alto, CA, USA), 1× non-essential amino acids (NEAA) (Gibco®, Palo Alto, CA, USA) and 50 IU penicillin/streptomycin (Sigma-Aldrich, Louis, MO, USA). Huh7 cells were incubated at 37 °C ± 5% $CO_2$ and were sub-cultured or cryopreserved as they reached 80% confluency. The genotype of CHIKV used in this study was the East/Central/South African genotype (accession number: MY/065/08/FN295485). The virus was propagated in Huh7 cells maintained in the working media consisting of RPMI-1640 medium supplemented with 2% FBS, 1× NEAA, and 50 IU

penicillin/streptomycin. The supernatants were harvested, titrated using plaque assay, and kept at −80 °C for further use.

## 2.2. Flavonoid and Agonists

The bioflavonoid compound used in this study, fisetin, was purchased from INDOFINE Chemical Company (Hillsborough, NJ, USA). Fisetin was dissolved in dimethyl sulfoxide (DMSO; Sigma Aldrich, Louis, MO, USA) to a stock concentration of 50 mM and kept at −20 °C for further use. When needed, fisetin was diluted to the working concentration of 30 µM in serum-free RPMI-1640 and filtered through a syringe filter with a 0.2 µm pore size (Millipore, MA, USA). The agonists used were the synthetic lipid analog, CRX-527 at 0.01 µg/mL (InvivoGen, San Diego, CA, USA), and the imidazoquinoline amine analog to guanosine, imiquimod-R837 at 5 µg/mL (InvivoGen, San Diego, CA, USA) for the induction of TLR4 and TLR7 pathways, respectively.

## 2.3. Virus Plaque Assay

A monolayer of Huh7 cells was cultured in a 48-well plate (Corning®, Steuben County, NY, USA). Upon infection, the viral supernatants were diluted (1:10) in the working media. The infection inoculum was added, and the plate was incubated for 1 h at 37 °C ± 5% $CO_2$. After 1 h, the inoculum was removed from each well, and the plaque media consisting of RPMI-1640 medium supplemented with 2% FBS, 1× NEAA and 50 IU penicillin/streptomycin, and 0.8% high viscosity carboxymethyl cellulose (HV-CMC; Sigma-Aldrich, Louis, Louis, MO, USA); was added to each well. The plate was then incubated for 48 h at 37 °C ± 5% $CO_2$. After 48 h, the plaque media was removed from each well, and the wells were washed with 1× phosphate-buffer saline (PBS). After washing, the cells were fixed with 4% paraformaldehyde (PFA) for 30 min at 4 °C. The virus plaques were stained by adding 0.5% crystal violet after washing the fixed cells with 1× PBS.

## 2.4. RNA Extraction, Antiviral Genes, and Viral Yield Reduction Assay

Viral RNA from the assay supernatants and cell lysates were extracted using QI-Aamp® Viral RNA Mini Kit (Qiagen, Hilden, Germany) and RNeasy Mini Kit (Qiagen, Hilden, Germany), respectively, following the manufacturer's protocol. The eluted viral RNA was then amplified and quantified in the virus yield assay using quantitative real-time polymerase chain reaction (qRT-PCR). The primers targeting the E1 gene of CHIKV as well as human glyceraldehyde 3-phosphate dehydrogenase (GAPDH), interferon-stimulated gene 15 (ISG-15), MX dynamin-like GTPase 2 (MX-2), 2'-5'- oligoadenylate synthetase 3 (OAS-3), and protein kinase R (PKR) were purchased from Integrated DNA Technologies (San Diego, CA, USA) and are listed in Supplemental Table S1. The qRT-PCR was performed using Step-OnePlus Real-Time PCR System (Applied Biosystems, Foster, CA, USA) with SensiFAST™ SYBR® Hi-ROX One-Step Kit (Bioline Meridian Bioscience, London, UK) following the manufacturer's protocol and using serially diluted standards of known CHIKV concentrations. Relative mRNA expression ($2^{-\Delta\Delta Ct}$ method) of the human antiviral genes was determined by normalization to GAPDH expression (Livak and Schmittgen., 2001). The temperature cycling parameters were 45 °C for 10 min, 95 °C for 2 min and 40 cycles of 95 °C for 5 s, 60 °C for 10 s and 72 °C for 5 s. The amplified cDNA products were verified by melting curve analysis.

## 2.5. Cytotoxicity Assay

A monolayer of Huh7 cells was cultured in a 96-well plate (Corning®, Steuben County, NY, USA). Fisetin was serially diluted (1:2) in the working media in the range of 25 µM to 800 µM upon treatment on the Huh7 cells. The agonists, CRX-527 (0.01 µg/mL to 0.16 µg/mL) and imiquimod-R837 (0.62 µg/mL to 10.0 µg/mL) were serially-diluted (1:2) in working media upon treatment on the Huh7 cells. The treated cells were incubated at 37 °C ± 5% $CO_2$. The CellTiter 96® Aqueous One Solution Cell Proliferation Assay (MTS;

Promega, WA, USA) was added to the cells following the manufacturer's protocol after 48 h of incubation. The absorbance was determined using a plate reader Infinite 200 PRO (Tecan, Zurich, Switzerland) and Magellan™ software v7.2 (Tecan, Zurich, Switzerland). The half-maximal cytotoxic concentration ($CC_{50}$) and maximum non-toxic dose (MNTD) of fisetin on Huh7 cells at 48 h were determined using the GraphPad Prism 5 software (GraphPad Software, Inc., San Diego, CA, USA).

### 2.6. Pre-Treatment and Post-Infection Treatment Assay

Expression of TLR proteins associated with TLR4 and TLR7 pathways, as well as pro- and anti-inflammatory markers, were monitored upon pre-treatment of Huh7 cells with the respective TLR agonists and infection with CHIKV (MOI = 1). The viral factor was monitored through virus yield assay, plaque assay, and immunofluorescence assay. Depending on the downstream assays, a Huh7 cells monolayer was cultured in a 24-well plate (Corning®, Steuben County, NY, USA) or 25 cm² flask (Corning®, Steuben County, NY, USA) in maintenance media. The cells were induced with the TLR4 agonist CRX-527 at 0.01 µg/mL (InvivoGen, San Diego, CA, USA) and TLR7 agonist imiquimod-R837 at 5 µg/mL (InvivoGen, San Diego, CA, USA) overnight at 37 °C ± 5% $CO_2$. Following the treatment, the agonist was removed, and the cells were infected with CHIKV (MOI = 1) and incubated for 1 h at 37 °C ± 5% $CO_2$. The infectious inoculum was removed, replaced with the working media, and incubated at 37 °C ± 5% $CO_2$ for 24 or 48 h. In the post-infection treatment assay, a monolayer of Huh7 cells was cultured in a 24-well plate (Corning®, Steuben County, NY, USA) or 25 cm² flask (Corning®, Steuben County, NY, USA) in maintenance media; depending on the purpose of the downstream assays. The inoculum, CHIKV with a multiplicity of infection (MOI = 1), was added to the cells, and the plate or flask was then incubated at 37 °C ± 5% $CO_2$ for 1 h. After 1 h, the inoculum was removed, and 30 µM fisetin was added to the cells. The plate or flask was then incubated for 24 or 48 h following which the supernatant of the cells was recovered and the cell lysates were prepared. After the incubation time in both assays, the morphology of the cells was viewed using bright-field microscopy, and the assay supernatants and cell lysates were collected to perform downstream assays. The supernatants were used to perform plaque and virus yield assays, while the cell lysates were used to perform immunoblot analysis. The half-maximal inhibitory concentration ($IC_{50}$) was determined using the GraphPad Prism 5 software (GraphPad Software, Inc., San Diego, CA, USA).

### 2.7. Immunofluorescence Assay (IFA)

Treated and infected Huh7 cells from the assays above were washed with 1× PBS and fixed with 4% PFA for 30 min at 4 °C. After fixation, the cells were stained for TLR4 (rabbit polyclonal to TLR4; Abcam, MA, USA), TLR7 (rabbit polyclonal to TLR7; Abcam), and anti-E2 (mouse monoclonal to E2; Novus Biologicals, Colorado, CO, USA) antibody respectively. After overnight incubation at 4 °C, the cells were washed with 1× PBS before adding secondary antibodies, goat-anti-rabbit conjugated with Alexa Fluor® 488 (Abcam, UK) and goat-anti mouse conjugated with Alexa Fluor® 594 (Abcam, UK) and incubated for 1 h at 4 °C. The nuclei were stained with DAPI (Thermo Fisher Scientific, Waltham, OH, USA) for 1 h at 4 °C, the cells were washed with 1× PBS. The images were captured by Leica DMI6000B (Leica Microsystems, Wetzlar, Germany), and the immunofluorescence signal was measured and analyzed with the Leica DFC365FX (Leica Microsystems, Wetzlar, Germany) and ImageJ processing program (National Institute of Health, NIH, Bethesda, MD, USA).

### 2.8. Immunoblot Assay

The immunoblot assay was performed as previously described in Priya et al., 2013 [38]. The PVDF membrane was incubated with anti-IRF3 (rabbit polyclonal to IRF3; Abcam), anti-IRF3 pS396 (rabbit polyclonal to phosphorylated-IRF3 S396; Abcam), anti-

NF-κB p65 (rabbit polyclonal to NF-κB; Abcam), anti-NF-κB pS529 (rabbit monoclonal to phosphorylated-NF-κB S529; Abcam), anti-IRF7 (rabbit polyclonal to IRF7; Abcam) and anti-IRF7 pS477 (rabbit polyclonal to phosphorylated-IRF7 S477; Invitrogen, CA, USA). Rabbit polyclonal to GAPDH (Abcam) was used as the loading control for all samples. After overnight incubation, the membrane was washed and incubated at room temperature with horseradish peroxidase (HRP)-conjugated secondary antibodies, goat anti-mouse (Cell Signaling Technology, Danvers, MA, USA) or goat-anti-rabbit (Cell Signaling Technology,Danvers, MA, USA). The membrane was developed using Clarity™ Western ECL substrate (Bio-Rad Laboratories, Hercules, CA, USA) and was viewed using Gel Doc XR+ Molecular Imager (Bio-Rad Laboratories, Hercules, CA, USA) and the Image Lab 3.0 software (Bio-Rad Laboratories, Hercules, CA, USA). The image was further analyzed using the ImageJ processing program (National Institute of Health, NIH, Bethesda, MD, USA).

### 2.9. Cytokine Array

Human pro-and anti-inflammatory cytokines and chemokines at 48 hpi (post-infection) were quantitated using the Quantibody® Human Cytokine Array Q1 Kit (RayBiotech, Peachtree Corners, GA, USA) following the manufacturer's protocol. The array allows the determination of multiple cytokines concentration such as IL-1$\alpha$, IL-1$\beta$, IL-2, IL-4, IL-5, IL-6, IL-8 (CXCL8), IL-10, IL-12, IL-13, GM-CSF, GRO, IFN-$\gamma$, MCP-1 (CCL2), MIP-1$\alpha$ (CCL3), MIP-1$\beta$ (CCL4), MMP-9, RANTES (CCL5), TNF-$\alpha$ and VEGF. The fluorescence signals were scanned with Cy3-equivalent wavelength using Innoscan 710 series fluorescence microarray scanner (Innopsys Inc., Chicago, IL, USA) and then analyzed using the Q-analyzer software (RayBiotech, Peachtree Corners, GA, USA).

### 2.10. Statistical Analysis

The statistical analysis, such as non-linear regression curve, one-way analysis of variance (ANOVA), and Dunnett's post-test, was performed using the GraphPad Prism 5 software (GraphPad Software, San Diego, CA, USA). The data presented were representative of three independent experiments, and values were expressed as mean ± standard deviation (SD).

## 3. Results

### 3.1. Fisetin Reduced CHIKV Replication in a Dose-Dependent Manner

In order to ensure the concentrations used are non-cytotoxic towards Huh7 cells, the CC50 and MNTD of fisetin and the agonists in Huh7 cells at 48 hpi were determined (Supplemental Figure S1). Concentrations used in this study; 30 μM fisetin, 0.01 μg/mL CRX-527 and 5 μg/mL imiquimod-R837 were non-toxic to Huh7 cells as all concentrations used were below their MNTD; 1.22 mM, 8.99 μg/mL and 14.32 μg/mL, respectively. In the preliminary result (Supplemental Figure S2), fisetin inhibited CHIKV replication in a dose-dependent manner with the IC50 of 5.59 μM. The IC50 of CRX-527 and imiquimod-R837 were 6.51 ng/mL and 1.81 μg/mL, respectively (Supplemental Figure S3). Following the preliminary results (Supplemental Figures S2 and S3), fisetin at 30 μM was used in this study as it reduced CHIKV-E1 RNA copy number by ~1000-fold. 0.01 μg/mL CRX-527 and 5 μg/mL imiquimod-R837 was used in this study as they reduced CHIKV-E1 RNA copy number by 59.12% and 57.36%, respectively. The fisetin treatment was performed as a post-infection treatment since it is the most effective treatment type as reported by Lani et al., 2016 [10].

### 3.2. CHIKV Inhibitory Potential of Fisetin

As fixed concentrations for treatments were already determined, the ability of fisetin and agonists in reducing CHIKV replication was evaluated. RNA samples from assay supernatants and cell lysates were harvested. From the assay supernatants, 30 μM fisetin

reduced CHIKV-E1 RNA copy number by ~1000-fold at both time points to approximately $5.60 \times 10^2$ at 24 hpi and approximately $3.05 \times 10^3$ at 48 hpi when compared to mock (positive control) Huh7 cells (Figure 1A). In cell lysates, fisetin reduced CHIKV-E1 RNA copy number by ~100-fold to approximately $3.40 \times 10^3$ at 24 hpi and approximately $7.56 \times 10^4$ at 48 hpi (Figure 1B). Only ~10-fold CHIKV-E1 RNA copy number reduction was observed in cell lysates with CRX-527 ($4.19 \times 10^4$) and imiquimod-R837 treatment ($3.05 \times 10^4$), respectively (Figure 1B). Focusing on the effect of the treatments on CHIKV progeny dissemination in Huh7 cells, CHIKV plaque counts were $1.65 \times 10^3$ pfu/mL at 24 hpi and $3.15 \times 10^4$ pfu/mL at 48 hpi (Figure 1C, Supplemental Figure S4). Fisetin significantly reduced CHIKV plaque counts by ~100-fold pfu/mL, at both time points with plaque count of $5.00 \times 10$ pfu/mL at 24 hpi and $5.00 \times 10^2$ pfu/mL at 48 hpi (Figure 1C). CRX-527 only significantly reduced CHIKV plaque count at 24 hpi to $1.35 \times 10^3$ pfu/mL (Figure 1C, Supplemental Figure S4). Imiquimod-R837 only significantly reduced CHIKV plaque count at 48 hpi to $2.92 \times 10^4$ pfu/mL (Figure 1C, Supplemental Figure S4). The visualization of CHIKV-E2 protein in mock Huh7 cells resulted in the MFI ratio of $0.37 \pm 0.005$ at 24 hpi and $0.85 \pm 0.005$ at 48 hpi. The expression of CHIKV-E2 protein in fisetin-treated CHIKV-infected Huh7 cells was significantly reduced at both time points to the MFI ratio of $0.02 \pm 0.005$ at 24 hpi and $0.08 \pm 0.005$ at 48 hpi (Figure 1D, Supplemental Figure S5). CRX-527 significantly reduced CHIKV-E2 protein expression to the MFI ratio of $0.26 \pm 0.005$ at 24 hpi , but not 48 hpi with the MFI ratio of $0.86 \pm 0.005$ (Figure 1D, Supplemental Figure S5). Imiquimod-R837 significantly increased CHIKV-E2 protein expression at 24 hpi with the MFI ratio of $0.42 \pm 0.005$ and significant decrease in the protein expression was observed at 48 hpi with the MFI ratio of $0.35 \pm 0.005$ (Figure 1D, Supplemental Figure S5).

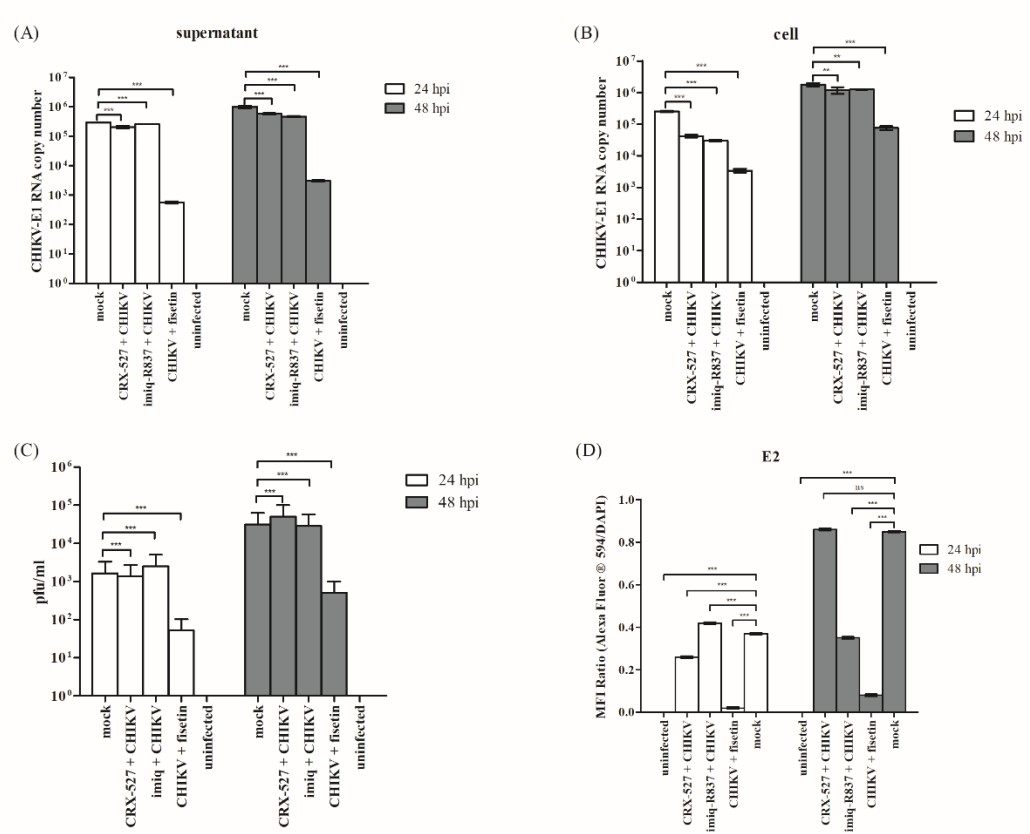

**Figure 1.** CHIKV inhibitory potential of fisetin and the TLR agonists. Viral yield reduction data from (**A**) assay supernatants; (**B**) cell lysates; (**C**) plaque count (pfu/mL) obtained from mock, agonist-treated, and fisetin-treated Huh7 cells (dilution 1:10); and (**D**) the mean fluorescence intensity (MFI) ratio of CHIKV-E2 protein in relation to DAPI. CHIKV infection was performed using MOI = 1. Data are representative of three independent experiments and values are expressed as mean ± SD. One-

way ANOVA ($p < 0.0001$) and Dunnett's multiple comparison post-test (* is $p < 0.01$ and *** is $p < 0.001$)) were performed by setting mock Huh7 cells of respective time points as an independent control.

### 3.3. Fisetin Reduced CHIKV-Induced Cytopathic Effects (CPE) in Huh7 Cells

The bright-field microscopic examination was performed concerning the effect of the treatments on CHIKV-induced CPE in Huh7 cells. CHIKV-induced CPE in Huh7 cells was manifested as cell shrinkage, detached, and convoluted cell extension, the increase of cell membrane protrusion, and plasma membrane blebbing (Figure 2). At 24 hpi, 40% of Huh7 cells showed CHIKV-induced CPE following the pre-treatment with 0.01 µg/mL CRX-527 (Figure 2A). More than 90% of Huh7 cells exhibited CHIKV-induced CPE at 48 hpi of the same treatment (Figure 2B). Pre-treatment with 5 µg/mL imiquimod-R837 demonstrated 10% of CHIKV-induced CPE after 24 hpi, and 80% at 48 hpi. Only 5% of CHIKV-induced CPE was observed in Huh7 cells treated with 30 µM fisetin at both time points (Figure 2).

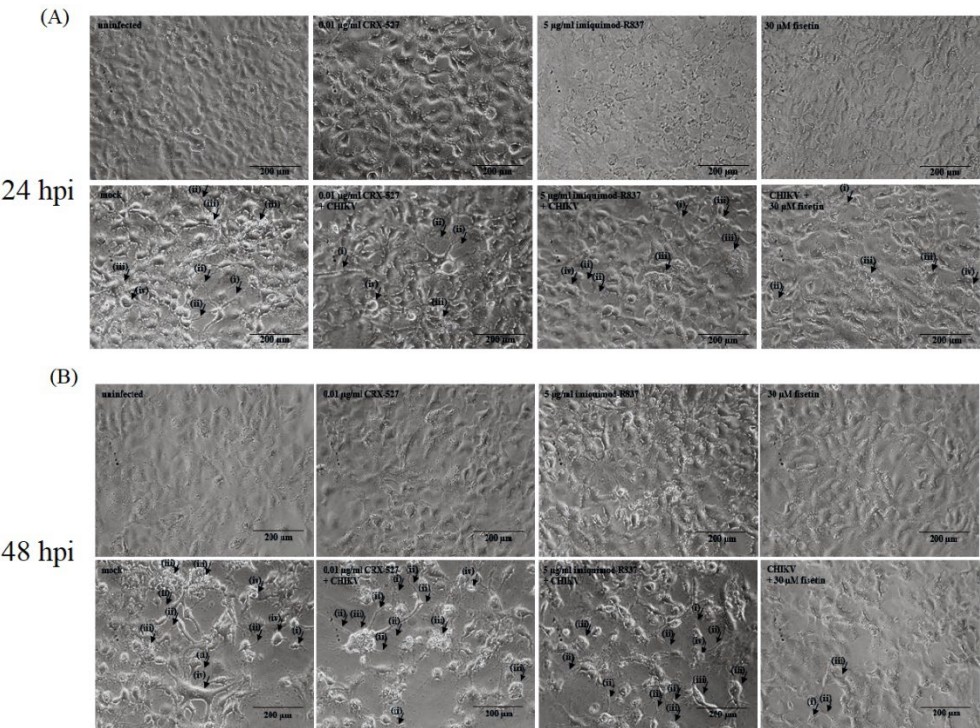

**Figure 2.** Bright-field microscopic examination (at 20×) of CHIKV-induced morphological changes. (**A**) 24 hpi and (**B**) 48 hpi in order to the left of the row; uninfected, CRX-527-treated, imiquimod-R837-treated, and fisetin-treated, with the top column being uninfected Huh7 cells and bottom as mock Huh7 cells. CHIKV infection was performed using MOI = 1. Labeled in the figures are: (i) cell shrinkage; (ii) detached and convoluted extension; (iii) increase of cell membrane protrusion; and (iv) plasma membrane blebbing.

### 3.4. Fisetin Treatment and CHIKV Infection Induced Endogenous TLR4 and TLR7 Protein Expression

The MFI ratio of TLR4 expression in uninfected Huh7 cells at 24 h was $0.240 \pm 0.005$ and $0.350 \pm 0.005$ at 48 h (Figure 3A, Supplemental Figure S5). Infection of CHIKV increased TLR4 expression in Huh7 cells with the MFI ratio of $0.570 \pm 0.005$ at 24 hpi and $0.710 \pm 0.005$ at 48 hpi. CRX-527 elevated TLR4 expression in mock Huh7 cells to the MFI ratio of $0.390 \pm 0.005$ at 24 hpi and $0.670 \pm 0.005$ at 48 hpi, but the MFI ratios remained lower than in mock Huh7 cells at both time points. Fisetin treatment resulted in a significantly higher MFI ratio at 24 hpi ($0.650 \pm 0.005$) and a non-significantly higher MFI ratio at 48 hpi ($0.720 \pm 0.005$), as compared to that of mock Huh7 cells (Figure 3A, Supplemental

Figure S5). The MFI of TLR7 expression in uninfected Huh7 cells at 24 h was 0.190 ± 0.005 and 0.450 ± 0.005 at 48 h (Figure 3B, Supplemental Figure S5). CHIKV significantly elevated TLR7 expression in Huh7 cells with the MFI ratio of 0.620 ± 0.005 at 24 hpi and 0.810 ± 0.005 at 48 hpi. Imiquimod-R837 increased TLR7 expression to the MFI ratio of 0.470 ± 0.005 at 24 hpi and 0.580 ± 0.005 at 48 hpi, although the MFI ratios remained lower than in mock Huh7 cells at both time points. Fisetin induced a significantly higher TLR7 MFI ratio at 24 hpi (0.640 ± 0.005) and at 48 hpi (0.830 ± 0.005), as compared to that of mock Huh7 cells (Figure 3B, Supplemental Figure S5).

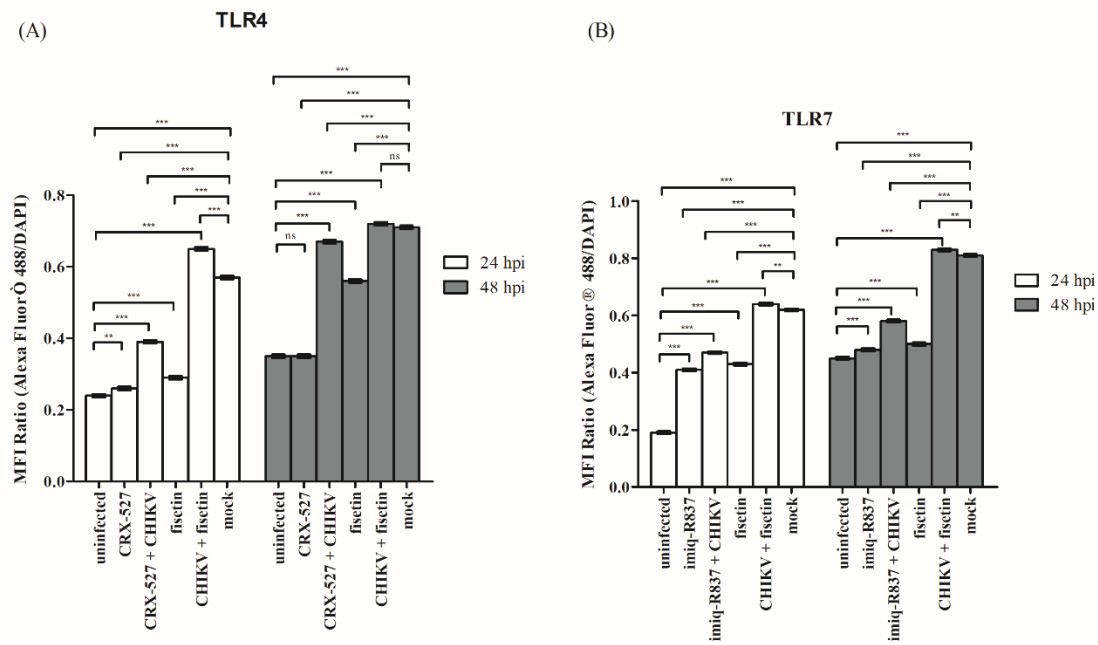

**Figure 3.** The mean fluorescence intensity (MFI) ratio of endogenous TLR4 and TLR7 protein in relation to DAPI. Data (**A**) TLR4 MFI ratio and (**B**) TLR7 MFI ratio are representative of three independent experiments and values are expressed as mean ± SD. CHIKV infection was performed using MOI = 1. One-way ANOVA ($p < 0.0001$) and Dunnett's multiple comparison post-test (** is $p < 0.01$ and *** is $p < 0.001$) were performed by setting uninfected and mock Huh7 cells as an independent control, respectively.

### 3.5. Induction of Endogenous Antiviral Genes in Huh7 Cells

Transcription and translation of antiviral genes are a response to viral infection. Following CHIKV infection, the mRNA expression level of antiviral genes was significantly elevated at both time points compared to uninfected Huh7 cells (Figure 4). CHIKV infection increased the antiviral genes mRNA level at 24 hpi to 1.75 ± 0.08-fold for ISG-15; 1.82 ± 0.25-fold for PKR; 2.59 ± 0.06-fold for MX-2 and 2.66 ± 0.20-fold for OAS-3 gene (Figure 4A–D). The elevation remained consistent at 48 hpi to 16.69 ± 0.52-fold for ISG-15; 3.16 ± 0.06-fold for MX-2 and 3.43 ± 0.20-fold for OAS-gene (Figure 4A,C,D); except a slight reduction at 48 hpi to 1.65 ± 0.01-fold for PKR gene (Figure 4B). Treatment of CHIKV-infected Huh7 cells with fisetin increased the antiviral genes mRNA level at 24 hpi to 2.66 ± 0.03-fold for ISG-15; 1.72 ± 0.11-fold for PKR; 6.27 ± 0.05-fold for MX-2 and 8.98 ± 0.06-fold for AS-3 gene. At 48 hpi, fisetin consistently increased the antiviral mRNA level to 16.69 ± 0.52-fold for ISG-15; 3.81 ± 0.43-fold for PKR; 7.89 ± 0.22-fold for MX-2 and 37.84 ± 0.07-fold for OAS-3 gene (Figure 4A–D). Only ISG-15 gene was elevated by 16.61 ± 0.54-fold upon pre-treatment of Huh7 cells with CRX-527 prior to CHIKV infection, surpassing that of mock Huh7 cells (Figure 4A).

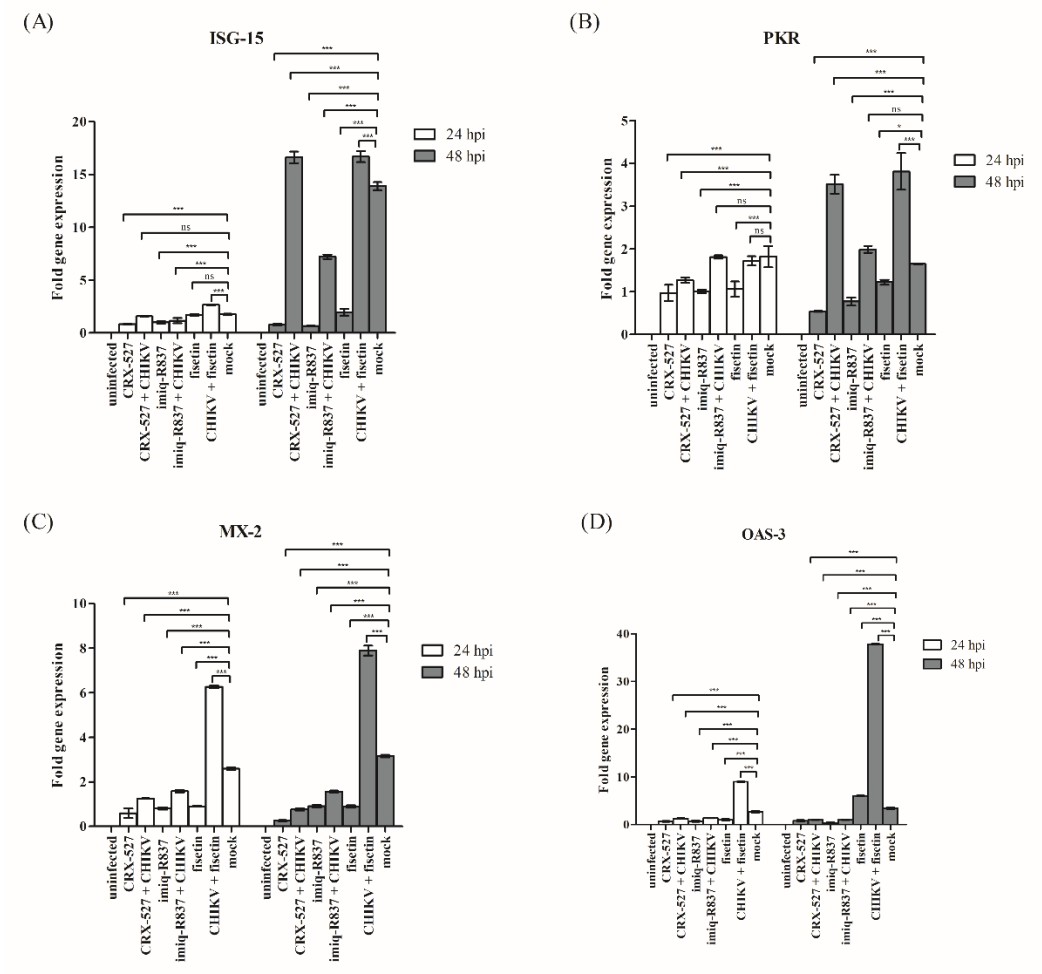

**Figure 4.** Gene expression profile of antiviral genes. (**A**) ISG-15, (**B**) PKR, (**C**) MX-2, and (**D**) OAS-3 in CHIKV-infected Huh7 cells at 24 hpi and 48 hpi. The data are normalized to human GAPDH mRNA and are expressed as a relative fold increase over normalized RNA from uninfected ($2^{-\Delta\Delta Ct}$ method). CHIKV infection was performed using MOI = 1. Data are representative of three independent experiments and values are expressed as mean ± SD. One-way ANOVA ($p < 0.0001$) and Dunnett multiple comparison post-test (* is $p < 0.05$, and *** is $p < 0.001$) were performed by setting uninfected and mock Huh7 cells of respective time points as an independent control.

### 3.6. Fisetin Reversed the Regulation of Pro- and Anti-Inflammatory Cytokines Evoked by CHIKV Infection

As one of the interests of this study is to highlight fisetin antiviral capacity in dampening CHIKV-induced inflammatory response, the level of pro- and anti-inflammatory cytokines were measured. The level of pro-and anti-inflammatory cytokines following pre-treatment with 0.01 µg/mL CRX-527 and 5 µg/mL imiquimod-R837, CHIKV infection (mock), and post-treatment with 30 µM fisetin at 48 hpi was presented in picograms per millilitre (pg/mL), as shown in Figure 5. From the results presented in Figure 5, fisetin reversed the effects of CHIKV infection on Huh7 cells' pro- and anti-inflammatory levels. Fisetin significantly decreased the level of all the pro-inflammatory cytokine that was elevated by CHIKV infection, which included IL-1α, IL-1β, IL-5, IL-6, IL-8, GM-CSF, GRO, IFN-γ, MCP-1, MIP-1α, MIP-1β, MMP-9, RANTES, TNF-α, and VEGF. Fisetin also significantly increased the level of the anti-inflammatory cytokine that was reduced considerably by CHIKV infection, such as IL-2, IL-4, IL-10, IL-12, and IL-13. The level of some cytokines such as IL-1β Figure 5B), IL-13 (Figure 5J), GRO (Figure 5L), MCP-1 (Figure 5N), and TNF-α (Figure 5S) of mock Huh7 cells treated with fisetin were comparable or non-significant to those of uninfected cytokines.

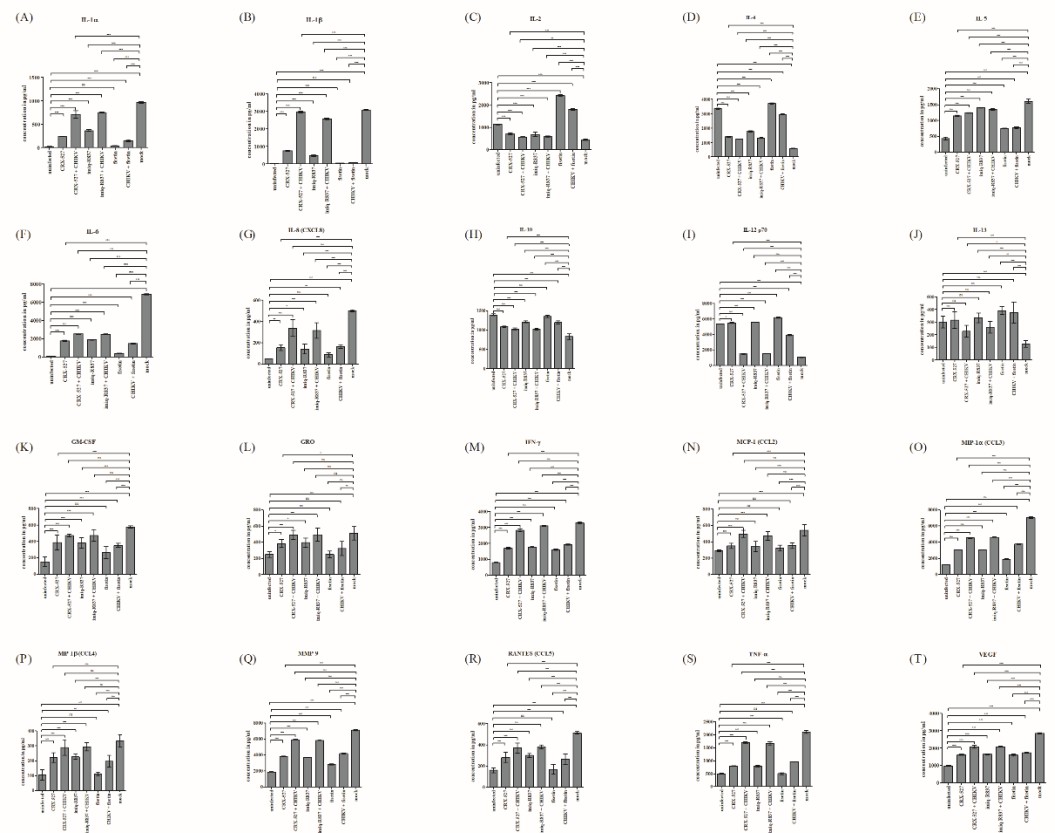

**Figure 5.** The cytokines and chemokines level in pg/mL. (**A**) IL-1$\alpha$, (**B**) IL-1$\beta$, (**C**) IL-2, (**D**) IL-4, (**E**) IL-5, (**F**) IL-6, (**G**) IL-8, (**H**) IL-10, (**I**) IL-12, (**J**) IL-13, (**K**) GM-CSF, (**L**) GRO, (**M**) IFN-$\gamma$, (**N**) MCP-1, (**O**) MIP-1$\alpha$, (**P**) MIP-1$\beta$, (**Q**) MMP-9, (**R**) RANTES, (**S**) TNF-$\alpha$, and (**T**) VEGF at 48 hpi. The concentration of chemokines/cytokines was interpolated from the standard curve of respective chemokines/cytokines standards. CHIKV infection was performed using MOI = 1. Data are representative of quadruplicates and values are expressed as mean ± SD. One-way ANOVA ($p < 0.0001$) and Dunnett multiple comparison post-test (* is $p < 0.05$, ** is $p < 0.01$ and *** is $p < 0.001$) were performed by setting uninfected and mock Huh7 cells as an independent control.

### 3.7. Effect of Fisetin on the Expression of Endogenous IRF3, IRF7 and Their Respective Phosphorylated Form in CHIKV-Infected Huh7 Cells

The effect of the treatments on mock Huh7 cells on the expression of the key transcription factors of antiviral and TLR-mediated inflammatory pathways was observed. Immunoblot results confirmed that CHIKV infection increased the expression level of endogenous p65 subunit of NF-κB and phosphorylated p65 subunit of NF-κB (S529) (Figures 6A,B,E and 7A,B,E) when compared to that of uninfected at both time points. Fisetin reduced the expression of p65 subunit of NF-κB and its phosphorylated form; pNF-κB(S529), at both time points in CHIKV-infected Huh7 cells, was superior to that treated with CRX-527 or imiquimod-R837 (Figures 6A,B,E and 7A,B,E). The expression level of endogenous IRF3 was increased at 48 hpi following CHIKV infection. The phosphorylated IRF3 (S396) expression, however, was decreased at both time points following CHIKV infection when compared to uninfected. Fisetin increased pIRF3(S396) expression in mock Huh7 cells exceeding the expression level in uninfected cells (Figure 6C–E). CHIKV infection significantly reduced the expression level of endogenous IRF7 and its phosphorylated form; pIRF7(S477) at both time points compared to that of uninfected cells. Fisetin elevated the expression of both proteins at both time points in mock Huh7 cells, surpassing the expression levels in uninfected cells, except for IRF7 at 24 hpi (Figure 7C–E).

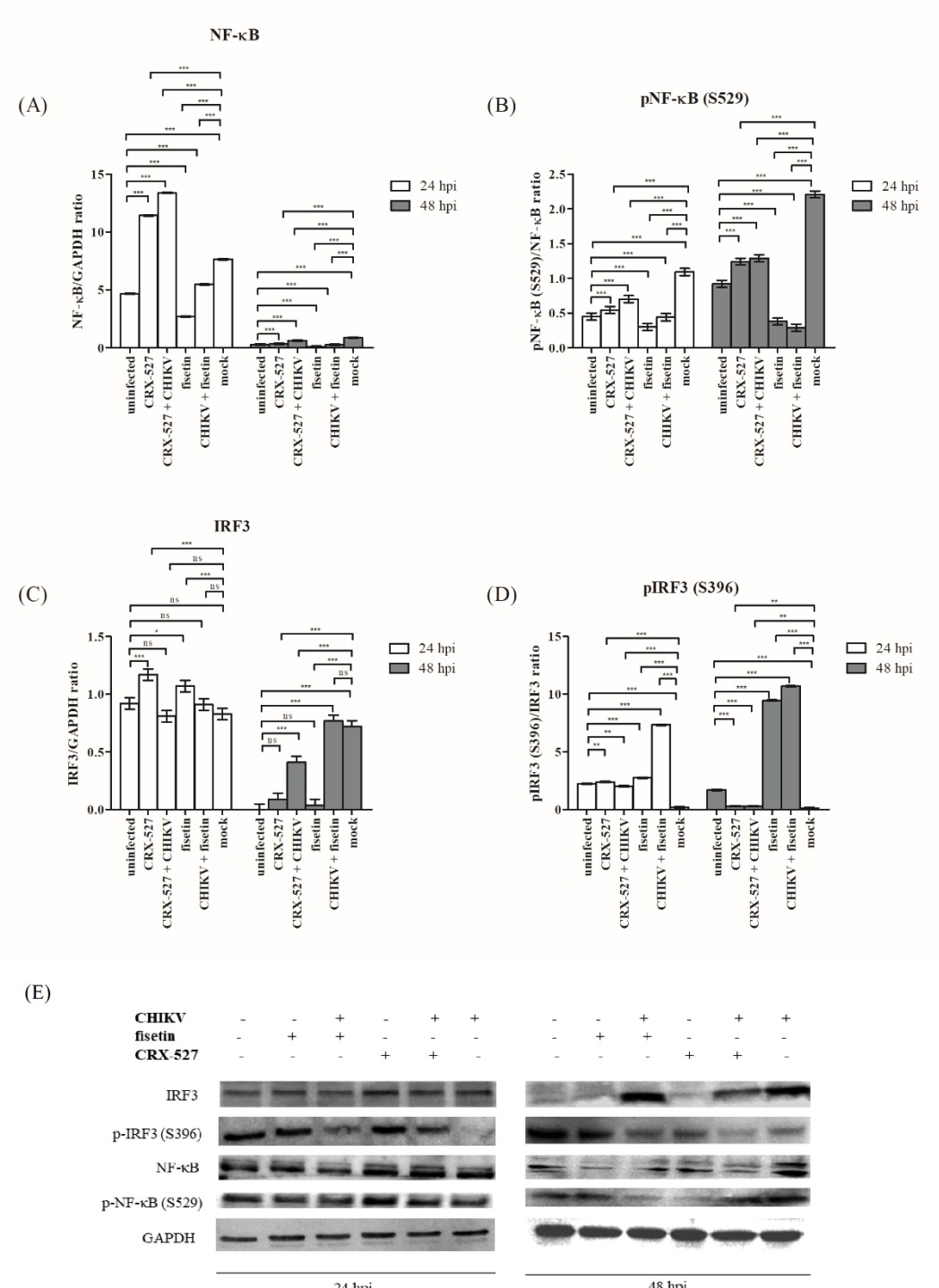

**Figure 6.** The expression of the transcription factors in the TLR4 pathway. Both (**A**) NF-κB and (**C**) IRF3 are analyzed relative to GAPDH, while (**B**) pNF-κB (S529) is relative to NF-κB expression, and (**D**) pIRF3 (S396) relative to IRF3. The blots were as in (**E**). CHIKV infection was performed using MOI = 1. Data are representative of triplicates and values are expressed as mean ± SD. One-way ANOVA ($p < 0.0001$) and Dunnett multiple comparison post-test (* is $p < 0.05$, ** is $p < 0.01$ and *** is $p < 0.001$) were performed by setting uninfected and mock Huh7 cells of respective time point as an independent control.

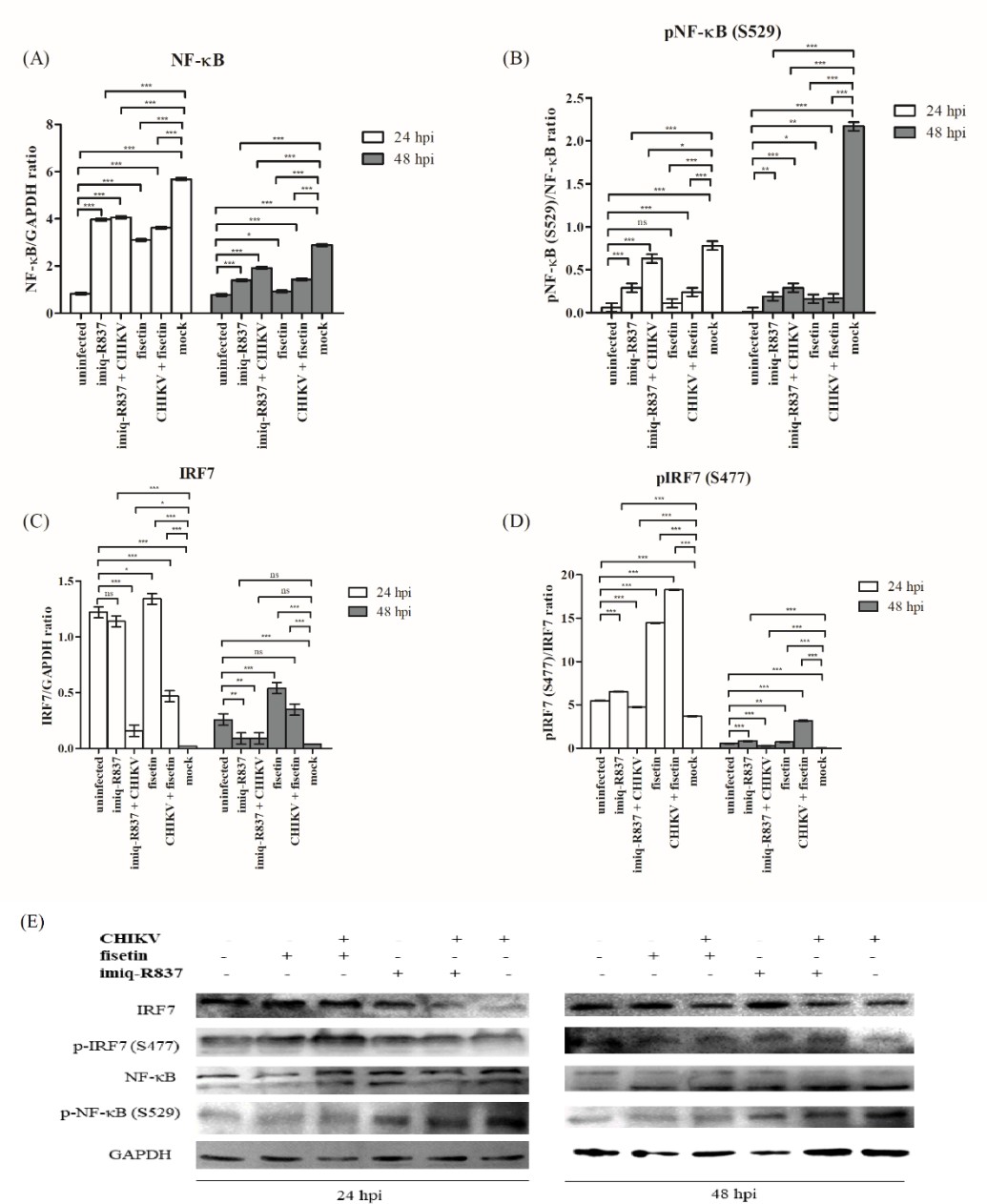

**Figure 7.** The expression of the transcription factors in the TLR7 pathway. Both (**A**) NF-κB and (**C**) IRF7 are analyzed relative to GAPDH, while (**B**) pNF-κB (S529) is relative to NF-κB expression, and (**D**) pIRF7 (S477) in relative to IRF7. The blots were as in (**E**). CHIKV infection was performed using MOI = 1. Data are representative of triplicates and values are expressed as mean ± SD. One-way ANOVA ($p < 0.0001$) and Dunnett multiple comparison post-test (* is $p < 0.05$, ** is $p < 0.01$ and *** is $p < 0.001$) were performed by setting uninfected and mock Huh7 cells of respective time point as an independent control.

## 4. Discussion

Arthritis in chronic CHIKV infection is caused by the influx of pro-inflammatory cytokines that disproportionates the homeostasis axis [40]. Inefficient CHIKV clearance leads to the persistency of CHIKV infection and aggravates the severity of the disease [41]. The TLR-mediated innate antiviral response is among the first responders in viral infection and is directly associated with activating inflammatory and antiviral responses [29]. The TLR-mediated innate antiviral response was reported in several previous studies and

was targeted for therapeutic strategy in anti-CHIKV drug development [38,42–44]. In this current study, we consolidated the involvement of TLR in CHIKV infection.

In agreement with the reported findings [39,45], our results suggested CHIKV envelope and single-stranded RNA (ssRNA) are the ligands of TLR4 and TLR7 receptors, respectively, as the expression of both receptors increased when Huh7 cells were infected with CHIKV. On the whole, CHIKV infection in Huh7 cells caused an increase in ISG-15, PKR, MX-2, and OAS-3 gene expression. ISG-15 was the most induced effector and it might be attributed to ISG-15 being the most highly responsive to type I IFNs [46–48] released from the downstream of activated TLR signaling pathways. Fisetin efficiency in elevating the level of ISG-15 gene expression was directly proportional to time. The ISG-15 gene expression elevation suggested that fisetin promotes the ISGylation of viral proteins [49] that disrupt CHIKV replication and at the same time, fisetin indirectly modulates IFN-γ production through ISG-15 stimulation [29]. Fisetin's ability in elevating PKR gene expression implied that fisetin induces an antiviral response by inhibiting mRNA translation, provoking apoptosis, and augmenting IFN response through the act of PKR [50,51].

Fisetin provided a better antiviral response in CHIKV-infected Huh7 cells with an increase in MX-2 and OAS-3 gene expression compared to CHIKV-infected Huh7 cells. These findings suggested that fisetin directs the host antiviral response, such as MxA monomers, to bind to viral nucleocapsids or other viral components to trap and degrade them [52]. The MX-2 gene expression elevation suggested that fisetin promoted viral nuclear intermediate dsDNA accumulation and altered viral capsid probably through binding of MX-2 amino-terminal domain [52]. Fisetin might also facilitate the binding of 2′,5′-oligoadenylates (product of OAS-3 gene) to RNase L triggers cleave of viral RNAs [53]. The involvement of these antiviral genes in CHIKV clearance is predicted as togaviruses are susceptible to the activity of these effector proteins. The results from antiviral gene expression are in agreement with the results from cell morphology observation, CHIKV-E1 RNA copy number, pfu/mL, and CHIKV-E2 protein expression.

Intense cytopathic effects and multifold viral loads in CHIKV-infected Huh7 cells revealed successful CHIKV replication that might be supported by the disproportion of pro- and anti-inflammatory cytokines, as shown by this study. The imbalance of the pro- and anti-inflammatory cytokines in CHIKV-infected Huh7 cells seems to be contributed by the ability of CHIKV to increase the expression of the prototypical pro-inflammatory cytokine, NF-κB, and to over-phosphorylate NF-κB. This study also found that CHIKV suppressed the expression of antiviral key transcription factors such as IRF3 and IRF7 and their phosphorylated forms, which hampered antiviral gene expression.

These results are consistent with other published findings with more engagements of TH2 and Treg responses associated with CHIKV infection. For example, IL-1β, IL-6, IL-8, GM-CSF, IFN-γ, and RANTES, are strongly induced in the acute phase of CHIKV infection [54] and are markedly increased in the CHIKV-infected group, with severe pain [55]. Hence, these cytokines are likely associated with CHIKV disease severity [56]. The GRO level is high in CHIKV-infected children [57], while IL-6, MCP-1, and TNF-α levels are high in neuro-CHIKV-infected cases [58]. The level of MIP-1α, MIP-1β, and VEGF are high in CHIKV-infected patients with chronic arthropathy as a response to tissue destruction [59]. A low level of IL-12 is detected in CHIKV-infected patients compared to the healthy control group [58]. Low IL-2, IL-4, and IL-13 during acute CHIKV infection are predictive markers for chronic joint pain [60], while higher IL-10 level is observed in recovered patients than in both acute and chronic patients [61]. This indicates that a sufficient level of anti-inflammatory cytokines such as IL-2, IL-4, IL-10, IL-12, and IL-13, is crucial for a better outcome of CHIKV disease.

Activating TLR4 and TLR7 with their respective agonists amplified their signaling pathways. However, the agonists failed to effectively regulate controlled inflammation and CHIKV clearance. Although pre-treating Huh7 cells with CRX-527 and imiquimod-R837 significantly increased endogenous TLR4 expression level at 24 hpi and TLR7

expression at both time points, respectively, both proteins expression was still significantly low in comparison to mock when the agonist-treated Huh7 cells were infected with CHIKV. Nonetheless, limitations such as TLR agonists concentration might have contributed to the lesser response of the downstream key proteins to compensate for the effect of negative regulators inflicted by CHIKV infection.

This might be due to the nature of the short half-life of TLRs and their downstream components, their location, and to what extent the receptor will reach a plateau after being exhausted by ligand-receptor interaction. TLRs' half-life is varied according to cell type and induction that are inflicted upon them. For example, the half-life of TLR4 in the alveolar macrophages derived from sham animals is 168 ± 32 min and is shortened by ~69% when treated with lipopolysaccharides [62]. The half-life of TLR7 in Huh7 cells is ~7 h and is shortened by Hepatitis C virus replication by half [63].

Unlike the action of TLR4 and TLR7 agonists, in this study, fisetin provided early yet prolonged protection against CHIKV infection and its cytopathic effects in CHIKV-infected Huh7 cells. Additionally, fisetin selectively modulates the expression of the pro- and anti-inflammatory cytokines, the key transcription factors, and the antiviral genes. As fisetin is already known for its antiviral activity, this study strengthens the fact that fisetin also modulates TLR-mediated innate antiviral response.

Fisetin reversed the effects of CHIKV infection on Huh7 cells' pro-and anti-inflammatory cytokines. Fisetin significantly increased anti-inflammatory cytokines and significantly reduced pro-inflammatory cytokines compared to positive controls, the CHIKV-infected Huh7 cells. Fisetin possibly restored the balance of these cytokines in CHIKV-infected Huh7 cells to the extent that the level of IL-1β, IL-13, GRO, MCP-1, and TNF-$\alpha$ was indifferent to the mock. As a high level of pro-inflammatory cytokines and a low level of anti-inflammatory cytokines are predictive markers for chronic joint pain and severe CHIKV, the balancing act provided by fisetin could offer better outcomes in CHIKV disease resolution.

Fisetin selectively increased the phosphorylation of IRF3 and IRF7 and eventually promoted the transcription of all four antiviral genes. These four distinguished interferon-inducible antiviral effectors are essential in the antiviral response, although they do not represent the entire repertoire of antiviral effectors [64]. They distinctly hinder viral transcription, degrade viral RNA, interrupt translation, or alter the proteasome to manipulate all steps of viral replication [64]. This may help explains how early viral clearance was achieved with fisetin treatment in CHIKV-infected Huh7 cells. Thus, there is a lesser requirement for increasing pro-inflammatory cytokines. The reduction of anti-inflammatory cytokines caused by CHIKV was overcome by treatment with fisetin. This balancing act brought about by fisetin in CHIKV-infected Huh7 cells was due to fisetin's ability to regulate the key transcription factors NF-κB, IRF3, and IRF7, which were previously manipulated by CHIKV infection.

## 5. Conclusions

The findings of this study suggest that CHIKV induces TLR-mediated antiviral responses. CHIKV reduced the expression and phosphorylation of IRF3 and IRF7. Hence, the expression of interferon-inducible antiviral effector genes, such as ISG-15, PKR, MX-2, and OAS-3, was affected. The increase in the expression of NF-κB, a prototypical pro-inflammatory key transcription factor, stimulates the production of pro-inflammatory cytokines resulting in the CHIKV-induced cytopathic effects. Pre-treatment of Huh7 cells with TLR4 and TLR7 agonists prior to CHIKV infection resulted in transient and moderate antiviral responses when compared to fisetin-induced antiviral activity. This study suggests that fisetin promotes TLR-mediated antiviral responses to limit CPE in CHIKV-infected Huh7 cells. Accordingly, fisetin promotes early and prolonged viral clearance by increasing the expression and phosphorylation of IRF3 and IRF7. As a result, the expression of antiviral effectors is increased, resulting in antiviral responses that inhibit CHIKV replication.

**Supplementary Materials:** The following supporting information can be downloaded at: https://www.mdpi.com/article/10.3390/immuno2040043/s1, Figure S1: Cytotoxicity assay of (A) fisetin, (B) CRX-527, and (C) imiquimod-R837 at 48 hpi on Huh7 cells; Figure S2: Viral yield assay represented as (A) CHIKV-E1 RNA copy number and (B) percentage reduction of CHIKV-E1 RNA copy number, at 48 hpi of treatment with fisetin; Figure S3: Viral yield assay represented as (A,C) CHIKV-E1 RNA copy number and (B,D) percentage reduction of CHIKV-E1 RNA copy number, at 48 hpi of treatment with CRX-527 and imiquimod-R837; Figure S4: CHIKV inhibitory potential of fisetin and the TLR agonists; Figure S5: Endogenous TLR4 and TLR7 induction in Huh7 cells with respective treatments and CHIKV infection; Table S1: Forward and reverse primers used for qRT-PCR.

**Author Contributions:** conceptualization, R.L. and S.A.; methodology, R.L. and P.H.; validation, R.L., B.-T.T., and S.-S.S.; formal analysis, R.L. and S.-S.S.; investigation, R.L.; resources, S.A. and S.-S.S.; data curation, R.L. and P.H.; writing—original draft preparation, R.L.; writing—review and editing, R.L., S.A., and P.H.; supervision, B.-T.T. and S.A.; project administration, S.-S.S.; funding acquisition, S.A. and S.-S.S. All authors have read and agreed to the published version of the manuscript.

**Funding:** This research was funded by the Ministry of Higher Education under the Fundamental Research Grant Scheme (FRGS/1/2019/SKK11/UM/02/3); and for niche research under the Higher Institution Centre of Excellence (HICoE) program; Project Code: MO002-2019. The authors have no other relevant affiliations or financial involvement with any organization or entity with a financial interest in or financial conflict with the subject matter or materials discussed in the manuscript apart from those disclosed. No writing assistance was utilized in the production of this manuscript.

**Institutional Review Board Statement:** Not applicable.

**Informed Consent Statement:** Not applicable.

**Data Availability Statement:** Not applicable.

**Acknowledgments:** We would like to acknowledge the staff of Tropical Infectious Diseases Research and Education Centre (TIDREC) and the Department of Medical Microbiology, Faculty of Medicine, Universiti Malaya, for their unwavering support throughout this study.

**Conflicts of Interest:** The authors declare no conflicts of interest.

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
