# Peer review of "Fisetin Modulates Toll-like Receptor-Mediated Innate Antiviral Response in Chikungunya Virus-Infected Hepatocellular Carcinoma Huh7 Cells"

_2673-5601, doi:10.3390/immuno2040043_

Round 1
Reviewer 1 Report
The authors (Rafidah Lani and colleagues) aimed to investigate the role of fisetin in chikungunya virus (CHIKV)-infected innate immunity in hepatocellular carcinoma Huh7 cell line. Their endpoint conclusion is that fisetin modulates TLR-mediated antiviral responses. The outcomes occur by affecting excessive inflammatory responses by thwarting over NF-κB phosphorylation and suppressing cytopathic effects in CHIKV-infected cells. Also, they reported that fisetin induces antiviral genes at an early time-point by promoting IRF3 and IRF7 phosphorylation. Although the manuscript is well-organized, some new data and solid methodology, but the text should be revised to improve clarity of the writing. The authors should consider the following issues to improve the strength of this manuscript:
Comments:
- Marginal differences found in TLRs expression between the group of untreated and fisetin-treated CHIKV-infected Huh7 cells. Cytopathic inhibitory effect of fisetin can interrupt the actual results generated by immunofluorescence assay. Authors need to provide immunofluorescence images as a supplement figures. Also, suggesting analyzing TLRs expression by western blotting method.
- Does CHIKV inhibitory potential of fisetin is TLR(s)-dependent? If so, I ask authors to perform additional experiment(s) to determine if altered TLR(s) expression restores the inhibitory effects of fisetin on viral infection.
- As the type I IFN is the key drivers for antiviral genes expression and regulated by principal mediators IRF3 and IRF7 and acting downstream of TLRs [10.1146/annurev-immunol-032713-120156]. Therefore, I recommend to authors to investigate the role of fisetin in type I IFN regulation by CHIKV-infected Huh7 cells. Does type I IFN has any role in inhibitory effects of fisetin in CHIKV infection?
- Each of the result section directly started with the results they found. Many readers might expect some background of the actual aim(s) and purpose of those experiments.
- Figure 5 is not visible clearly. Size of the figures with front should be readjusted.
- In vitro need to be change to italic format. Genes name should be in appropriate pattern.
Reviewer 2 Report
Lani et. al. presented their manuscript where they evaluated the effect of fisetin on CHIKV replication and the modulation of the immune response. The manuscript is relevant to the field but requires improvement which I listed below.
MAJOR COMMENTS
RESULTS
General comments for Figures. It was difficult to evaluate some figures since they are too small to zoom in. More organization and better presentation will improve the manuscript quality, see below details.
Include MOI used in each set of experiments, and the pre-treatment or post-treatment conditions as well. All result sections should include a statement, a hypothesis, and an objective per figure at least.
Figure 1. Authors should consider presenting all 48 hpi data in Figure 1, while 24 hpi may be included in Supplementary material. Indicate the MOI used. Labels in “X” axis can be modified as follow, Uninfected instead of Mock, Mock instead of CHIKV and other bars should only include the compound/drug’s name. Fig 1C Plaque assay figure should include a proper label as well. In figure 1D, the authors present the mean fluorescence intensity (MFI) data, please include the respective pictures of those experiments. Authors are encouraged to include FACS analysis of those positive cells, if possible.
Figure 3. Where is the statement for the transition from the model of infection to TLR experiments? What is the hypothesis and the objective? Include the pictures of the Immunofluorescence assay. Also indicate conditions such MOI used here and pre-treatments/post-treatment.
Figure 4. Where is the statement for the transition from TLR experiments to ISG gene expression? What are the hypothesis and the objective? “Y” label no need to include Delta Delta CT label, similarly in the Fig 4. Legend please omit it, this information should be described in detail in the Materials and Methods section.
Figure 5. Where is the statement for the transition from ISG experiments to cytokines and chemokines expression? What are the hypothesis and the objective? Also indicate conditions such MOI used. The Figure should be designed for a full page or part of it relocated in the supplementary section.
Figure 6 & 7. What is the hypothesis and the objective?. Indicate the MOI. Also share for revision the full row data of Western Blot images.
Discussion. Include in this section, the ISGs (ISG-15, PKR, MX-2 and OAS-3) gene expression data.
Conclusion. Review and modify the writing for ISG-15, PKR, MX-2, “Hence, antiviral protein translation from antiviral effectors such as ISG-15, PKR, MX-2, and OAS-3 was affected”. The authors evaluated RNA expression, but no protein expression for the mentioned ISGs.
SUPPLEMENTARY SECTION
Would authors explain the difference in CHIKV-E1 RNA copy numbers in Supplemental Figure 2A shows higher than 106 for fisetin but in Supplemental Figure 3A and 3C was near to 105. Different MOI used? Different viral stock?, cell passage?, or settings were performed in parallel. If so include a comment.
MATERIALS AND METHODS
2.6 Pre-treatment and post-infection treatment assay. Did authors evaluate the effect of CHIKV infection with fisetin pre-incubation (1h or 2 h)? or only post-infection? If so, what was the outcome?, If not why they did not perform this condition?
MINOR COMMENTS
Other observations:
Line 220, correction for “MNTD; 1.22 mL” indicate the fisetin concentration.
Please modify hours post-infection (hpi) or (h.p.i.) instead of “hours PI”.
Reviewer 3 Report
In this manuscript, Lani et al. demonstrated the antiviral mechanisms of fisetin. In detail, fisetin treatment and CHIKV infection induced endogenous TLR4 and TLR7 protein expression and endogenous antiviral genes production but reversed the regulation of pro- and anti-inflammatory cytokines evoked by CHIKV infection in Huh7 cells. Overall, it can be a valuable contribution to the field. However, several points require attention and should be addressed as described below.
1. It seems that supplemental materials are not provided. Please check.
2. Some figures are not shown clearly, such as Figure 1, 2, 5. Please re-arrange. Particularly, remove plaque image into supplement materials from Figure 1.
3. As shown in the title, Fisetin modulates toll-like receptor-mediated innate antiviral response in chikungunya virus-infected hepatocellular carcinoma Huh7 cells. How to establish the connection between TLR and innate antiviral response? Although the authors detected the gene expression of antiviral genes (ISG15, PKR, MX2 and OAS3), cytokines and chemokines, and the phosphorylation of IRF3, 7 and NF-κB, it is not enough to conclude that as other pathway may contribute to that such as RLR (RIG-I-like receptors).
4. In Figure 6E (right panel), why IRF3 is not detectable in mock-infected and mock-treated cells?
5. Please define *, ** and *** when statistical analysis was performed.
6. Please search the whole text for some minor errors to be corrected.
For example,
25cm2 flask in line 154
at 48h PI (post-infection) in line 202, hpi is good.
approximately 5.60 x 102 in line 230
Round 2
Reviewer 2 Report
Thanks for the authors' efforts to improve the manuscript. The authors responded most of this reviewer’s comments. However, there are still major observations.
Importantly, Supplement Figure 5A at 24 hpi, TLR4 staining, “uninfected” and “CRX-527” are duplicated images.
The statement: Lines 294 and 295, “Fisetin treatment resulted in a significantly higher MFI ratio at 24 hpi (0.650 ± 0.005)”, is not supported by the data presented in images supplemental Figure 5A, top panel at 24 hpi, CHIKV+fisetin”. This reviewer encourages the authors to present a WB data for TLR4 and TLR7 detection using the protein lysates from Figures 6 & 7.
Authors did not present the Western Blot raw data (no-cropped images) requested by this reviewer.
Line 289 and 290. Correction, indicate only h (hour) instead of h PI or hpi due these points correspond to uninfected cells.
Please comment in the discussion section the limited level of induction in TLR4 expression in CRX-527 uninfected treated cells.
Author Response
Thanks for the authors' efforts to improve the manuscript. The authors responded most of this reviewer’s comments. However, there are still major observations.
- Importantly, Supplement Figure 5A at 24 hpi, TLR4 staining, “uninfected” and “CRX-527” are duplicated images.
Thank you for noticing this. The image for “uninfected” in Supplement Figure 5A (24 hpi) is now changed to a new one.
- The statement: Lines 294 and 295, “Fisetin treatment resulted in a significantly higher MFI ratio at 24 hpi (0.650 ± 0.005)”, is not supported by the data presented in images supplemental Figure 5A, top panel at 24 hpi, CHIKV+fisetin”. This reviewer encourages the authors to present a WB data for TLR4 and TLR7 detection using the protein in lysates from Figures 6 & 7.
The image for “CHIKV + fisetin” supplemental Figure 5A (top panel at 24 hpi) is now changed to a new image that reflects the stated result. As we have mentioned in the previous round of revision to address reviewer No.1 concern, we believe that measuring TLRs expression by quantitative immunofluorescence method is much more reliable than the semi-quantitative western blotting method.
- Authors did not present the Western Blot raw data (no-cropped images) requested by this reviewer.
Unfortunately, the only images that we were able to retrieve from our central facilities were the analyzed images. We can assure you that the images were genuine and were only subjected to allowed modifications.
- Line 289 and 290. Correction, indicate only h (hour) instead of h PI or hpi due to these points corresponding to uninfected cells.
Corrections were made throughout the manuscript for this concern.
- Please comment in the discussion section on the limited level of induction in TLR4 expression in CRX-527 uninfected treated cells.
Necessary changes were made to this concern in the discussion section.
Reviewer 3 Report
Thanks for the authors' efforts to improve the manuscript. The reviewer has no further concerns.
Author Response
Thank you for reviewing and sharing inputs to improve our manuscript. The English language and style were verified by our native-English speaker colleague and, the minor spelling check was performed.
Round 3
Reviewer 2 Report
The authors responded to most of this reviewer’s comments. This reviewer has no further observations, except the notes below that may help to improve the manuscript.
All the best and thank you for the effort.
Please double-check carefully ALL statistic calculus for significant bars with *** placed on each Figure, for example, it is hard to see if there is a “real” ***p<0.001 on the statistical analysis bars in different Figures:
· Fig 1A, mock versus CRX-527 + CHIKV at 24 hpi;
· Fig 1C, mock versus CRX-527 + CHIKV at 24 hpi;
· Fig 6A NK-kB, uninfected versus CRX-527 at 48 hpi;
· Fig 6B pNF-kB, uninfected versus CRX-527 at 24 hpi;
· Fig. 6D pIRF3, uninfected versus CRX-527 at 24 hpi.
Please correct the Fig. 4D OAS-3 in which the statistical analysis bars are moved to the right in the 24 hpi plot.
Finally, in Line 302, the statement “Fisetin induced a significantly higher TLR7 MFI ratio at 24 hpi (0.640 ± 0.005)” is not supported by the image (picture) CHIKV + fisetin at 24 hpi in Supplemental Figure 5B.
Author Response
Thank you for your constructive comment. For the *** on the bar graph in the mentioned figures 1A, 1C, 6A, 6B, and 6D; we have no problem viewing the statistical annotations clearly. The statistical analysis bars in Fig 4D are now moved to the left (correct position) as requested. The image for CHIKV + fisetin in supplemental figure 5B (24 hpi) was also changed to reflect its written result.